# Challenges associated with e-cigarette use by people in custody in Scottish prisons: a qualitative interview study with prison staff

Rachel O'Donnell ,[1] Ashley Brown ,[1] Douglas Eadie,[1] Danielle Mitchell,[1] Linda Bauld ,[2] Evangelia Demou ,[3] Richard Purves ,[1] Helen Sweeting ,[3] Kate Hunt [1]

RO'D and AB contributed equally.

RO'D and AB are joint first authors.

[1]Institute for Social Marketing and Health, Faculty of Health Sciences and Sport, University of Stirling, Stirling, UK
[2]Usher Institute and SPECTRUM Consortium, College of Medicine and Veterinary Medicine, University of Edinburgh, Edinburgh, UK
[3]MRC/CSO Social & Public Health Sciences Unit, University of Glasgow, Glasgow, UK

**Correspondence to**
Dr Rachel O'Donnell;
r.c.odonnell@stir.ac.uk

## ABSTRACT

**Objectives** Little is known about the perspectives of staff working in prisons where e-cigarettes are permitted. Scotland now permits people in custody (PiC), but not staff/visitors to use e-cigarettes, following implementation of smoke-free prisons policy in 2018. Previous studies, conducted before and immediately after the introduction of e-cigarettes in Scottish prisons, have evidenced stakeholder support for their use by PiC. This study focuses on key challenges associated with e-cigarette use in prisons, using data collected from prison staff once e-cigarettes had been allowed in a smoke-free environment for 6–9 months.

**Setting** Five prisons in Scotland.

**Participants** Sixteen qualitative interviews were conducted with prison staff from five prisons varying by population (sex, age and sentence length). Data were managed and analysed using the framework approach.

**Results** While these staff confirmed strong support for the smoke-free prison policy and reported some benefits of replacing tobacco with e-cigarettes, they also spoke of the challenges e-cigarettes pose. These included: workplace e-cigarette vapour exposures; perceptions that e-cigarettes provide a new, effective way for some PiC to take illegal drugs, particularly new psychoactive substances; organisational challenges relating to the value attached to e-cigarettes in prisons; and implications for long-term nicotine use and tobacco cessation. Staff anticipated difficulties in tightening restrictions on e-cigarette use by PiC given its scale and significance among this population.

**Conclusions** Maximising the benefits of e-cigarette use by PiC is likely to require multiple measures to support effective and safe use and e-cigarette reduction/cessation where desired. This includes monitoring any misuse of e-cigarettes, and provision of guidance and support on appropriate e-cigarette use and how to limit or quit use if desired. Findings are relevant to jurisdictions considering or planning changes in prison smoking or vaping policies.

## Strengths and limitations of this study

► Conducting interviews with prison staff in a private room facilitated open discussions about their experiences of the challenges associated with e-cigarette use among people in custody in Scottish prisons.

► The use of a qualitative approach in this study broadens understanding of risks and challenges of use of e-cigarettes in this distinct setting and corresponding measures that might minimise harms and maximise benefits.

► While some staff groups (eg, female members of staff, current vapers) may not be adequately represented in the sample, the inclusion of staff from a diverse set of prisons provides confidence that the breadth of challenges associated with e-cigarettes in prisons are likely to be reflected.

## INTRODUCTION

In anticipation of the November 2018 implementation of the country's comprehensive (indoor/outdoor) smoke-free prison policy, legislative changes occurred which enabled people in custody (PiC) in Scotland to purchase and use single-use and rechargeable e-cigarettes in rooms (cells) and outdoor spaces from September 2018. The places where e-cigarette use is permitted mirrors the places where PiC were allowed to smoke tobacco prior to implementation of the smoke-free policy (ie, in designated rooms and in some outdoor spaces). Tobacco stopped being sold in prisons via the prison shop ('canteen') ~2 weeks before the smoke-free policy was introduced. Prison staff and visitors remained prohibited from smoking, or using e-cigarettes (hereafter vaping), on prison property. Information on e-cigarette products sold via the canteen in Scottish prisons and prison smoking cessation services is available elsewhere.[1 2] The introduction of e-cigarettes in prisons in Scotland (and in other jurisdictions such as England and Wales, and the USA[3 4]) has potentially important implications for the health and

safety of PiC and prison staff, and the operation of prison systems. There is very limited evidence on e-cigarette use in prisons, with no studies reporting on the perspectives of staff working in prisons where vaping is permitted among PiC, thus limiting understandings of its perceived benefits and risks in this distinct setting.

Current research evidence on e-cigarette use in prisons comes from two studies of tobacco and vaping in Scottish prisons (the Tobacco in Prisons study (TIP) and the E-cigarettes in Prisons study).[1 5–7] Evidence from TIPs, and responses to a public consultation on smoke-free prisons (by the Scottish Prison Service),[8] showed that e-cigarettes received broad support from stakeholders in anticipation of the introduction of smoke-free prison policy. For example, TIPs surveys in May–June 2018 (after the smoke-free policy announcement but before it was known e-cigarettes would be made available to PiC) and May–June 2019 (6-9 months after use of (recharge-able) e-cigarettes was first allowed and smoke-free policy was implemented in Scottish prisons) both showed the majority of prison staff (~75%) and almost all PiC (~90%) agreed that 'e-cigarettes should be available to help prisoners to stop smoking/manage without tobacco'.[5]

Complementary qualitative research suggests this broad support for e-cigarettes in Scottish prisons has been substantially influenced by perceptions that they are beneficial in helping PiC and staff to cope with the major operational change represented by the removal of tobacco.[9] Favourable perceptions of e-cigarettes in Scottish prisons also reflect broader evidence on their appeal and effectiveness for some current and ex-smokers (eg, those who feel unwilling or unable to quit nicotine)[10 11] and evidence on reduced harms to users and bystanders from vaping relative to smoking.[12]

However, important issues and challenges in relation to permitting PiC to vape were also voiced by prison staff and some PiC in anticipation of their introduction,[1 13] as well as by commentators in other jurisdictions.[14 15] These concerns relate to e-cigarette safety, implications for long-term tobacco and nicotine cessation (eg, uncertainties about whether e-cigarette use in prison increases or decreases the likelihood of return to smoking on release), and risks associated with vaping related to unique features of prison environments. These include: rules around earnings and purchases that mean PiC are not always able to (formally) buy (enough of) the items they want or 'need'; operation of illicit prison economies in which any sought-after item can be traded or 'bought' using alternative currencies; and well established practices of PiC repurposing items, including for illicit purposes such as drug taking.[1] The benefits of e-cigarettes in prisons are well covered in our previous publications (including focus groups with prison staff before rules were changed to allow PiC to vape[7 13]; interviews with PiC immediately after e-cigarettes were introduced[1] and again once new rules on e-cigarettes and smoking had bedded in[9]; and repeat cross-sectional surveys of prison staff and PiC[5]). This paper, therefore, focuses on the challenges of permitting the use of e-cigarettes in prisons identified by prison staff, based on novel interview data collected after vaping in prisons had become well established (8–11 months after sales of rechargeable e-cigarettes first began in Scottish prisons and 6–9 months after smoke-free rules were implemented). Together with our previous studies, the current findings will be relevant for prison systems (and institutions such as mental health inpatient services) seeking to implement, and maximise the benefits of, smoke-free policies.

## METHODS

### Patient and public involvement

The study was designed to ensure that the views of prison staff could be heard at different stages of the process of implementing smoke-free prisons in Scotland. A Research Advisory Group which included representation from prison management and union members provided feedback on the overall design of the study, study materials and early findings to inform data collection and interpretation. Staff within the prisons were involved in the dissemination of information about the study and recruitment to the study. Personnel involved in implementing and managing smoke-free policies were also involved in discussion of findings to establish recommendations for future policymakers and implementers.

### Sample and recruitment

Data were collected between May and August 2019 from 16 Scottish Prison Service staff working in 5 (of Scotland's 15) prisons which varied by prison population (sex, age and sentence length) and capacity. Participants were recruited through designated staff contacts who were asked to invite staff from a range of work roles (eg, residential; admissions; health and safety; prison management), to enable the exploration of diverse views. Our contacts scheduled one-to-one meetings between researchers and staff who expressed interest in participation. Sample characteristics are detailed in table 1.

### Data collection

Interviews (averaging 35 min) were conducted in a room at the prison which allowed the participant to speak freely. After providing information about the study and answering any questions, written or audiorecorded verbal consent was obtained from each participant by one of three interviewers (RO'D, AB and DE) before she/he commenced the interview. The topic guide covered: participant background, smoking and vaping history; views and experiences related to vaping in the prison context; and views on the positive and negative consequences of vaping by PiC for prison staff, the prison regime and PiC themselves. Question wording and order were varied as appropriate and participants were invited to raise any additional points they considered important.

### Data analysis

Fifteen of the 16 interviews were audiorecorded and transcribed by a professional transcription agency; detailed

**Table 1** Participant characteristics (n=16)

| Work role | | No of years of service | |
|---|---|---|---|
| Residential officer | 8 | 1–10 | 2 |
| Non-residential officer | 6 | 11–20 | 6 |
| Managerial | 1 | 21–30 | 5 |
| Missing | 1 | 31–40 | 2 |
| | | Missing | 1 |
| **Sex** | | **Age (years)** | |
| Male | 13 | 18–30 | 2 |
| Female | 3 | 31–40 | 2 |
| | | 41–50 | 3 |
| | | 51–60 | 8 |
| | | Missing | 1 |
| **Smoking status** | | **Vaping status** | |
| Non-smoker | 9 | Non vaper | 11 |
| Ex smoker | 4 | Ex vaper | 3 |
| Current smoker | 2 | Current vaper | 1 |
| Missing | 1 | Missing | 1 |

notes were made in lieu of audiorecording of one interview on participant request. All interview content was deidentified, then analysed (by RO'D, AB and DM) using the framework approach.[16] This approach is well suited to the analysis of in-depth interview data, as summarisation of data into a framework grid within the NVivo program facilitates analysis across cases without losing the wider context of each participant's account.[17] A thematic framework was developed (by AB and RO'D) to guide data management and analysis, based on deductive (considering the topic guide, relevant literature) and inductive (reading transcripts) techniques. In preparation for detailed analysis, data summaries were written in relevant cells of the framework grid (RO'D, DM and AB), incorporating hyperlinks to transcripts to facilitate data retrieval. RO'D reviewed all summaries to check data interpretation and consistency of approach. Data summaries were then used to identify high level themes (AB and RO'D) before further in-depth analysis was conducted (RO'D), focusing on data about challenges posed by e-cigarettes in prisons. RO'D and AB finalised themes based on re-examining data and reflexive team discussions. Themes are presented below alongside illustrative quotes in boxes 1–3; each quote indicates the prison the participant worked in, their interview ID number and smoking and vaping status.

## RESULTS

### Staff perceptions of the smoke-free prison policy and e-cigarette availability for PiC

Prior to reporting staff perspectives of the challenges of permitting the use of e-cigarettes in prisons, we briefly summarise, for context, their opinions on smoke-free

**Box 1   Staff perceptions of the smoke-free prison policy and e-cigarette availability**

1. 'It's night and day in that the smell of tobacco's not there. You're not breathing it in. …it's not in your clothes. You're not bringing it home to your family. That to me is a big thing.' *L4-NS-NV*
2. 'The fact that the e-cigarettes gave an alternative to the cigarettes actually made the transition of the jail changing over to no smoking a lot easier…it was a massive help to us.' *B1-NS-NV*
3. '…making jails smoke-free, we could just say, right that's it [(no nicotine substitute]), and we would have had bother [(from PiC]). But would we have got through that bother, and then not had the smokers [(vapers])?' *C2-NS-NV*
4. 'The vapes made the process from smoking tobacco to there not being tobacco, it made that an easy process because we gave the prisoners a new option, while we kind of said to ourselves…'Why not just take the hit [i.e., without introducing e-cigarettes] and see how it goes with no tobacco?' Would that not have been better? Especially because I don't think there's enough research into the e-cigarettes and the vaping as well. We don't know exactly what kind of consequences there are.' *O3-NS-NV*

policy and beliefs about the benefits of e-cigarette availability in prisons. Staff interviewed for this study voiced strong overall support for the smoke-free prison policy and reported positive impacts of removing tobacco from prisons, including benefits for their own health and comfort while at work (box 1-1).[1 13] E-cigarettes were generally believed to have supported smoke-free prison policy implementation by helping PiC to manage without tobacco (box 1-2). Benefits for reducing tobacco-related harms among PiC and staff were sometimes discussed. However, while most participants acknowledged some benefits associated with the introduction of e-cigarettes in prisons, several drawbacks were also discussed, leading some to reflect on whether smoke-free prisons could have been achieved without allowing PiC to vape. For example, some staff suggested in retrospect that the service might have been able to manage any temporary disruption that could have arisen from the removal of tobacco from prisons (box 1-3&4)

### Staff exposures to e-cigarette vapour

Several participants expressed doubts about the safety of vaping for themselves as bystanders (box 2a-1), in many cases referring to the absence of long-term studies on exposure to e-cigarette vapour (box 2a-2). Concerns about exposure to e-cigarette vapour while at work were discussed in the context of very widespread vaping among PiC (box 2a-3). In this context, some staff reported practical challenges in avoiding exposures to e-cigarette vapour when carrying out tasks such as cell searches (box 3a-4). One participant wondered whether the persistent cough she had started to experience was linked to workplace exposure to e-cigarette vapour.

### Misuse of e-cigarettes among PiC to take illegal drugs

Concerns about the risks associated with exposure to e-cigarette vapour were amplified because participants

---

### Box 2   Staff exposures to e-cigarette vapour, and misuse of e-cigarettes among people in custody to take illegal drugs

#### (2a) Staff exposures to e-cigarette vapour

1. 'I wonder what the harm is of passive 'smoking' with the vapes. I know we've taken away a lot of chemicals and tar and things that were in tobacco and that's great but…what is in the actual oils [e-liquids] and is that potentially doing harm? *C1-CS-EV*

2. Am I any better off? We won't know for 20 years.' *C2-NS-NV*

3. 'I would say ninety-five per cent of them use e-cigarettes, ninety percent or more…' *G2–NS-NV*

4. 'If you went to somebody's cell and it's a spontaneous job and somebody's vaping in the cell, how realistic is it for you to say…'You need to stop vaping! You need to stop that for an hour before I come into this cell'? It's impossible.' *O3-NS-NV*

#### (2b) Misuse of e-cigarettes to take illegal drugs

1. 'There has been a lot of instances of vapes being broken, or improvised, shall we say …prisoners now know they can put paper in. It's obviously soaked in NPS (novel psychoactive substances)…and that's now their hit.' *L3-NS-NV*

2. 'The thing that prisoners have got is time to figure out how to take drugs…not every prisoner is like that obviously. It's only a certain amount that smoke drugs…but they adapt a different way of doing it and the thing is, they've got time to figure that out.' *O3-NS-NV*

3. 'The vape's just good [(for NPS use)] because it's enclosed… and also how do you police 500, 400 and odd people, in some cases the bigger jails, you've got a thousand odd prisoners. How do you police every single person with a vape?' *B1-NS-NV*

4. 'A lot of staff become ill because they've entered somebody's cell to speak to them and by the time they've left they've started to feel… its [secondhand exposure to NPS] taken effect. So I'm quite conscious if I go to a cell. If it's smoky [filled with e-cigarette vapour], I'm not so keen to go in there…it feels like it's a matter of time given how many of my colleagues have been affected by it…Headaches, high blood pressure, heartrate zooming up, becoming shaky and unsteady. Staff have been really, really poorly with it.' *C1-CS-ExV*

5. 'It's a completely different workplace now, and this is definitely something new for me, in the years I've worked here I've never known this before.' *O4-ExS-ExV*

---

### Box 3   The value attached to e-cigarettes in prisons, and long term nicotine use and smoking cessation

#### (3a) The value attached to e-cigarettes in prisons

1. '…boys scratching the door, 'Oh boss, can you get us any vapes?' And then they're shouting to the other guys, 'Anyone got any?' and they were handed a capsule [e-liquid] over with wee dribbles in it to keep them going.' *C2-NV-NS*

2. 'You have to try and nip that in the bud [vulnerable PiC getting into e-cigarette related debts], if you catch it, straight away.' *G1-ExS-NV*

3. 'There are, obviously, other things that get the guys in debt, but this seems to be the quick way…to borrow one [(refill)], pay back two, [(and)] not have the money [(to buy your own)] the following week.' *G1-ExS-NV*

4. 'We've already had an incident whereby one of the girls thought that she had caught something from the misuse of the vapes and that immediately stopped within the hall, for as far as we can try and control it.' *B1-NS-NV*

5. 'We had one [(case)] this morning actually. Someone was screaming and shouting because her vape has been nicked.' *C1-CS-ExV*

#### (3b) Long-term nicotine use and maintained smoking cessation

1. 'I think the smoking cessation works well, but I think the motivation for [(some)] prisoners to stop 'smoking' disappears when they can have vapes. I think they were more motivated [(to quit smoking)] just before the ban.' *C3-NS-NV*

2. 'They've purchased these vapes, bought them in jail, the first time they've taken them out with them [(after release from prison)] they've kept a hold of them because they've not got the financial money to go and buy cigarettes.' *B1-NS-NV*

3. 'Very seldom do I come across a prisoner who wants to stop smoking. Those who are being released keep telling me they're going over to the shop across the road to buy a pack of cigarettes when they get out. So they've got no intention of…the vape is just a short term thing for a lot of them.' *G2-NS-NV*

#### (3c) Perceived potential solutions to the challenges raised by vaping in prisons

1. 'There's no way you can, you know, manipulate that [single-use] device to [(take NPS)]…and that's why a lot of the staff are asking that they bring that in.' *B1-NS-NV*

2. 'We've gone from smoking to vaping, are we just maintaining now, or do we now have to start looking at having a reduction programme for the vapers?' *C2-NV-NS*

3. 'Do the human rights of a prisoner outweigh the [(human rights of the)] guys who are working here?' *O3-NV-NS*

---

in all prisons viewed (rechargeable) e-cigarettes as providing a new and effective way for some PiC to take NPS (box 2b-1), particularly since some PiC had a lot of unoccupied time that could be spent on (devising) illicit activity (box 2b-2). While some noted the removal of lighters as part of smoke-free policy may have contributed to increased NPS use (replacing cannabis which PiC were less able to ignite), one reported that the number of PiC being placed on observations for drug use had increased over several years and so current NPS use might only reflect an underlying trend.

Staff working in residential areas of prisons gave accounts of how some PiC adapted e-cigarette devices to enable the ingestion of vapour from NPS-soaked paper and noted the unpredictable and adverse effects of this practice on PiC. Residential staff also noted that NPS use was easily concealed within an e-cigarette, and often odourless, adding to concerns about 'policing' NPS use (box 2b-3). They often spoke of feeling vulnerable at work and sometimes when driving home, due to the risks and effects of (unwitting) secondhand exposures to NPS, especially when entering cells. Accounts of harms to themselves or colleagues because of involuntary inhalation of NPS as a bystander were described (box 3b-4) and a few suggested the effects of NPS use in prison in general had significantly changed their work environment, for example, by undermining safety and security for staff and PiC (box 2b-5).

### Value attached to e-cigarettes in prisons

E-cigarettes were viewed as having become an integral part of prison life over a relatively short space of time,

---

serving similar functions to those previously fulfilled by tobacco, including as a valued commodity/form of prison currency. As with other 'alternative' currencies or valued items in prisons, this could create problems for staff and the prison system (as well as for PiC themselves).

First, staff reported having to manage problems, including tensions in the halls and challenging behaviour when PiC ran out of e-liquids and wanted to borrow e-cigarette products from others (box 3a-1). Borrowing often came at a price; it was reportedly not uncommon for PiC to have to repay over the amount borrowed, leading to debt and in some cases bullying, in particular of more vulnerable PiC (box 3a-2). One participant suggested that while issues of borrowing and debt (and bullying) were encountered in relation to other products, e-cigarette related debts could grow relatively quickly (box 3a-3). Borrowing tobacco (before the smoke-free policy was introduced) was seen as somewhat easier for PiC to manage, because it could be lent in small amounts, whereas borrowing e-cigarette products involved the loan of an e-liquid or device.

Second, the price of e-cigarette products was perceived to be a moderate or serious burden for some PiC, especially those on very low incomes, as tobacco had been for some PiC before the smoke-free rules. This sometimes led to the sharing of e-cigarette devices, and the exchange of part-used e-liquids (to enable use of different flavours at no additional cost) which raised concerns for staff about possible virus and disease transmission (box 3a-4). (Note: interviews were conducted over a year before the first documented COVID-19 cases.)

There were several other ways that e-cigarettes created additional challenges for staff; for example, when PiC misplaced their device, the onus was on staff to break from their duties to locate it, and when e-cigarettes were reported as stolen (box 3a-5). Several staff reported regular vaping in prohibited areas (eg, vaping outside designated rooms and some outdoor spaces), and while some described this as unintentional, others reported instances where PiC had deliberately hidden e-cigarettes while moving through areas of the prison where vaping was not allowed, or had refused to go to appointments unless they were able to take their e-cigarettes with them. Some staff were willing to put PiC on report for repeated breaches, but others viewed the policing of e-cigarettes as too difficult because non-compliance was commonplace.

## Long-term tobacco and nicotine use

Staff reported that prior to the introduction of e-cigarettes, some PiC had intended to become nicotine-free once smoke-free rules were implemented. However, few PiC had reportedly achieved this; most previous smokers were vaping e-liquids containing nicotine long after tobacco had been removed from prisons (box 3b-1). Staff were generally unsure about what support was available in prisons for PiC who wished to quit vaping and noted that e-liquids were only available in higher strengths in prisons, which some staff viewed as a potential barrier to enabling PiC to reduce use of e-cigarettes/nicotine dependence.

While staff noted that some individuals might continue to vape exclusively after leaving prison, for example, for financial reasons (box 3b-2), several relayed discussions with PiC who intended to smoke tobacco on release, and in some cases this was considered a fait accompli (box 3b-3), reflecting, for example, people's intentions to resume co-use of cannabis and tobacco.

## Perceived potential solutions to the challenges raised by vaping in prisons

Participants expressed varied views on solutions to the perceived challenges of vaping in prisons. These included only selling single use e-cigarettes in prisons to limit product misuse; and supporting PiC to manage and, if desired, reduce use of rechargeable e-cigarettes (box 3c-1). A few participants suggested e-cigarettes should be completely prohibited in prisons, with one suggesting this would better balance the 'rights' of staff with those of PiC (box 3c-2&3). However, staff acknowledged that greater vaping restrictions were likely to be unpopular with PiC, and very challenging to implement given the scale and significance of vaping in prisons among PiC.

## DISCUSSION

This study provides novel insights into prison staff experiences of the challenges associated with allowing e-cigarettes to be sold and used among PiC in smoke-free prisons, alongside further evidence of perceived individual and organisational benefits of e-cigarettes in prisons. Findings suggest that staff were concerned, to varying degrees, by perceived negative impacts of e-cigarettes on worker health and safety, job tasks, and work environment. Given that the roles of prison staff are highly demanding,[18] it is perhaps unsurprising that participants were very alert to potential risks associated with this major organisational change while generally expressing support, in national surveys, for having e-cigarettes in prisons for PiC.[5]

An area of concern to participants was the potential health impacts of (short and long term) exposure to e-cigarette vapour at work. These concerns are particularly understandable given that rates of vaping are around 10 times higher among PiC than the Scottish general population[19 20] and that prison staff need to enter rooms where PiC regularly vape, to provide urgent care or maintain security. Prisons appear unusual among UK public sector organisations in permitting use of e-cigarettes indoors; a recent study found that vaping was prohibited indoors in all National Health Service Trusts and almost all Higher Education Institutions.[21] Differences in vaping policies between prisons and other workplaces may be viewed particularly negatively by prison staff given that the UK prison workforce was exposed to secondhand smoke for over a decade longer than other UK workforces because prisons were partially exempt from Scotland's 2006 national smoke-free laws. Concerns among prison staff in Scotland about the potential health

effects of e-cigarettes on bystanders (and users) may also reflect broader anxieties about the safety of e-cigarettes; similar concerns have been expressed by other occupational groups for example, healthcare staff.[22] Among the general population, a US study found that participants generally perceived e-cigarette vapour exposures to be 'moderately harmful' to health; almost half believed that vaping should be prohibited in indoor public places such as restaurants.[23]

The use of e-cigarettes for illicit drug use is understandably of significant concern to prison staff given the adverse impacts that this can have on those living and working in prison and operational stability.[24] There is a growing body of evidence on the challenges and harms posed by NPS use in prisons, which had been identified as a challenge within the prisons several years prior to the introduction of smoke-free prison policy in Scotland.[25–27] Findings from TIPs interviews exploring post-implementation perspectives on Scotland's smoke-free prison policy among a separate sample of staff and PiC also suggest that the smoke-free prison policy may have contributed to changes in the use of NPS, specifically through misuse of e-cigarettes.[9] Understandings of the general risks posed by e-cigarettes for illicit drug taking, including NPS, are developing: a 2018 systematic review[28] suggested several potential areas of risk. These include: that e-cigarettes might be viewed as a 'safer and innocuous method to experiment and try drugs'; levels of illicit drug use might increase due to the ease with which e-cigarettes can be used for drug delivery; the emergence of new patterns of drug use; and illicit drug use may be easier to conceal, leading to increased problems with detection by authorities and greater risks of 'unintended or malicious' exposures to bystanders (p107). It will be important to monitor these risks, including in prison settings. These findings highlight the ongoing challenges for prison authorities to find appropriate e-cigarette devices for use in prisons, and other secure settings such as in-patient mental health services.

Prison staff also discussed challenges related to the value attached to e-cigarettes in prisons. These appeared similar to those previously caused by tobacco prior to smoke-free prison policy and are to be expected, given prison rules on purchasing items from the prison shop and the informal economy in prisons.[29] (Analysis of canteen spend will be reported separately.) Nonetheless, some staff might view such issues as particularly troublesome at a time when the workforce is facing wider challenges including increasing numbers of PiC, staff sickness absence rates and increased working hours.[30]

Finally, staff reported potential limitations or challenges of e-cigarettes for reducing tobacco harms in PiC long term. Discomfort with continued use of nicotine among PiC may stem from staff knowledge that some individuals have a strong desire to become 'free' of substance dependence, as well as from (mis)understandings of health risks of nicotine[31] or perceptions that someone who vapes has not truly stopped 'smoking'.[32]

Staff members' acknowledgement of risks that many PiC might resume smoking soon after leaving smoke-free prisons, despite having exclusively vaped for a period, reflects the substantial challenges that people often face in transitioning from prison to the outside world.[33] Prison staff did not raise concerns about use of e-cigarettes by the minority of never smokers in the prison population.[34] Ongoing monitoring would be helpful in tracking this issue in the future, so that swift corrective action could be taken if required.

Findings reported here will be of interest to jurisdictions planning on making their prison systems smoke-free and those considering use of e-cigarettes for harm reduction. They highlight the need for any policies on e-cigarette use in prisons to balance the needs of those PiC who wish to use e-cigarettes (in lieu of tobacco) against calls for precautionary measures to limit (or eliminate) e-cigarette vapour exposures because of a lack of long-term evidence on e-cigarette safety. Based on current evidence regarding their effects on air quality and health,[35] restricting indoor vaping in prisons to private rooms (cells) would appear proportionate, although there is a need for ongoing monitoring of the evidence and clear communication for the rationale for policies to prison staff. Like smoking previously, where allowed, vaping in prisons is likely to become normative and culturally embedded over time, potentially presenting future challenges for a prison service should the need to introduce tighter restrictions on e-cigarette use arise. Other challenges associated with the use of e-cigarettes among PiC might be reduced through providing guidance and support on appropriate use and on how to limit or quit use if desired. New practitioner guidance has been developed in Scotland subsequent to these staff interviews, to support PiC to reduce or cease use of e-cigarettes[36] and smoking relapse prevention interventions spanning the pre-post release period would be beneficial. Further research is planned to assess use and impact of this guidance, and to understand whether further measures are required to support behaviour change (eg, changes to the e-cigarettes that are sold in prison). Broader responses are also required to tackle illicit drug use among PiC, including measures to reduce environmental drivers of prison drug use.[37]

We believe the use of one-to-one interviews in this study is a strength because they enable in-depth exploration of people's perspectives and of confidential or sensitive issues.[38] Findings from this study complement and extend our previous work exploring staff perspectives on e-cigarettes using data gathered via focus groups before announcement of the smoke-free policy,[13] and work exploring perspectives among PiC post-implementation.[9] A key study limitation is that certain participant groups (eg, females, current vapers) may not be adequately represented in the sample, limiting our ability to explore potential between-group differences in perspectives. However, the inclusion of staff from a diverse set of prisons provides confidence that the breadth of challenges associated with e-cigarettes in prisons are likely to be reflected well in

the data. It is important to note that data were collected prior to the introduction of guidance for practitioners on how to support PiC to reduce or stop using e-cigarettes and before the COVID-19 pandemic, both of which may have important implications for vaping in prisons and are priorities for future studies.

In conclusion, this detailed analysis, focused on the challenges associated with vaping in prisons from the perspective of prison staff, identified five main issues: staff concerns about exposures to e-cigarette vapour at work; use of e-cigarettes for illicit drug taking among PiC; the value attached to, and trading of, e-cigarette products in prisons; and implications for long-term nicotine and tobacco use. Addressing these challenges is likely to require a combination of measures specific to e-cigarettes and broader measures to address (illicit) drug use and promote overall health.

**Acknowledgements** We are grateful to the prison staff who took part in the interviews, staff within the Scottish Prison Service who assisted with the study and facilitated access, and NHS staff who have supported our research on smoke-free prisons.

**Contributors** KH, AB, HS and LB developed the study, with input from RO'D, DE, RP and the SPS Research Advisory Group. AB and KH liaised with the prison service and reported back on ongoing findings. AB prepared the ethics applications. RO'D, AB and DE conducted the qualitative interviews; all are highly experienced qualitative researchers. AB, RO'D and DM conducted analysis using the Framework approach. RO'D, AB and KH drafted the manuscript and all revisions. All other authors read a subsample of the transcripts and contributed to interpretation of the data and review of the final manuscript. As guarantors, KH, AB and RO'D accept full responsibility for the conduct of the study, had access to the data, and controlled the decision to publish.

**Funding** This work was supported by Cancer Research UK (C45874/A27016). ED and HS were also supported by the UK Medical Research Council MC_UU_12017/12 and Scottish Chief Scientist Office SPHSU12.

**Competing interests** None declared.

**Patient and public involvement** Patients and/or the public were involved in the design, or conduct, or reporting, or dissemination plans of this research. Refer to the Methods section for further details.

**Patient consent for publication** Not applicable.

**Ethics approval** Ethical approval was granted by the SPS Research Access and Ethics Committee (approval confirmed 23.8.18; no ref number provided) and by the University of Stirling General University Ethics Panel (ref: GUEP 497). Participants gave informed consent to participate in the study before taking part.

**Provenance and peer review** Not commissioned; externally peer reviewed.

**Data availability statement** No data are available.

**ORCID iDs**
Rachel O'Donnell http://orcid.org/0000-0003-2713-1847
Ashley Brown http://orcid.org/00000002-2307-5916
Linda Bauld http://orcid.org/00000001-7411-4260
Evangelia Demou http://orcid.org/0000-0001-8616-525X
Richard Purves http://orcid.org/00000002-6527-0218
Helen Sweeting http://orcid.org/0000-0002-3321-5732
Kate Hunt http://orcid.org/0000-0002-5873-3632

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
