## [Reviewer comments · BMJ Open]

ARTICLE DETAILS

TITLE (PROVISIONAL)	The challenges associated with e-cigarette use by people in custody in Scottish prisons: a qualitative interview study with prison staff
AUTHORS	O'Donnell, Rachel; Brown, Ashley; Eadie, Douglas; Mitchell, Danielle; Bauld, Linda; Demou, Evangelia; Purves, Richard; Sweeting, Helen; Hunt, Kate

VERSION 1 – REVIEW

REVIEWER	Lin Xiao Chinese Center for Disease Control and Prevention, Tobacco Control Office
REVIEW RETURNED	10-May-2021

GENERAL COMMENTS	Comments to author: 1. This qualitative study described the challenges of e-cigarette use in prisons with established smoke-free policies, using data collected from prison staff. It sounds good. However, some evidences seems not very strong to support your findings. For example, in line 35 of page 11, you mentioned participants in all prisons viewed e-cigarette as providing a new and effective way for some PiC to take new psychoactive substances, based on the Table 3b-5, which was “The instances of prisoners being under the influence of NPS, has increased quite dramatically over the past couple of months.” This study was conducted 6-9 months after the smoke-free policies implementation. Why NPS increased quite dramatically over the past couple of months, rather than after the policy implementation? Did you explore the reasons? In line 52 of page 11, you mentioned residential staff gave account of how some PiC adapted e-cigarette devices to enable the ingestion of vapour from NPS-soaked paper. It is very important evidence to link misuse of e-cigarette use with illicit drug use. However, it wasn’t appeared in table 3b.2. In your paper, you mentioned the motivation for prisoners to stop smoking disappears when they can have vapes. And most previous smokers were vaping e-liquids containing nicotine long after tobacco had been removed from prisons. In addition, I wonder if there were some prisoners who did not smoking before, but started to use e-cigarette and became nicotine addiction, since e-cigarette was so popular in these prisons? By the way, I am little bit curious about the sentence in line 38 of page 13. Before the ban, the smokers couldn’t get any tobacco in prisons?3. In the results of this paper, you showed us perceived potential solutions to the challenges raised by vaping in prisons. It provides audiences important information for jurisdictions considering or planning changes in prison vaping policy. As you described in strengths of the study, this study broadens understanding of challenges of using e-cigarettes in this distinct setting and
---

	corresponding measures that might minimize harms and maximize benefits. Thus, you should mention them in the abstract. 4. Suggest displaying table 3 after the “Misuse of e-cigarette among PiC to take illegal drugs”, since “NPS” was first appeared in this paragraph.
--	---

REVIEWER	Kahlia McCausland Curtin University - Perth City Campus, Collaboration for Evidence, Research and Impact in Public Health, School of Public Health
REVIEW RETURNED	31-May-2021

GENERAL COMMENTS	Thank you for the opportunity to review this paper, it was an interesting read. Please find my comments in the attached PDF. The reviewer provided a marked copy with additional comments. Please contact the publisher for full details.
---

VERSION 1 – AUTHOR RESPONSE

Response to Reviewer #1:

1. This qualitative study described the challenges of e-cigarette use in prisons with established smoke-free policies, using data collected from prison staff. It sounds good. However, some evidences seems not very strong to support your findings. For example, in line 35 of page 11, you mentioned participants in all prisons viewed e-cigarette as providing a new and effective way for some PiC to take new psychoactive substances, based on the Table 3b-5, which was “The instances of prisoners being under the influence of NPS, has increased quite dramatically over the past couple of months.” This study was conducted 6-9 months after the smoke-free policies implementation. Why NPS increased quite dramatically over the past couple of months, rather than after the policy implementation? Did you explore the reasons? In line 52 of page 11, you mentioned residential staff gave account of how some PiC adapted e-cigarette devices to enable the ingestion of vapour from NPS-soaked paper. It is very important evidence to link misuse of e-cigarette use with illicit drug use. However, it wasn’t appeared in table 3b.

Our response: We have changed quote 3B-1 to better illustrate the finding that staff spoke about how some PiC were using e-cigarettes to take NPS. This finding is also reflected in our complementary Tobacco In Prisons study, which we have now referred to in the Discussion.

NPS use had been identified as a challenge within prisons several years prior to the introduction of smoke-free prison policy in Scotland, and we now clarify this in the Discussion section. Given time constraints during interviews, we were not able to explore with staff the full range of factors influencing trends in use of NPS in prisons over time. We agree this is an important area of future study.

2. In your paper, you mentioned the motivation for prisoners to stop smoking disappears when they can have vapes. And most previous smokers were vaping e-liquids containing nicotine long after tobacco had been removed from prisons. In addition, I wonder if there were some prisoners who did not smoking before, but started to use e-cigarette and became nicotine addiction, since e-cigarette was so popular in these prisons? By the way, I am little bit curious about the sentence in line 38 of page 13. Before the ban, the smokers couldn’t get any tobacco in prisons?

Our response: Your point about the use of e-cigarettes by people who did not smoke prior to the introduction of the smokefree policy is interesting, and whilst we don’t have any evidence that e-cigarettes have been taken up by non-smokers in custody, we believe this is an important area for future research to explore.

We have now clarified in the Introduction that tobacco stopped being sold in each prison canteen (shop) ~two weeks before the smokefree policy was introduced, to provide the context required for the quote on page 14, line 38.

3. In the results of this paper, you showed us perceived potential solutions to the challenges raised by vaping in prisons. It provides audiences important information for jurisdictions considering or planning changes in prison vaping policy. As you described in strengths of the study, this study broadens understanding of challenges of using e-cigarettes in this distinct setting and corresponding measures that might minimize harms and maximize benefits. Thus, you should mention them in the abstract.

Our response: Many thanks for your suggestion. We have now included details in the conclusion of the Abstract on measures that could be taken to minimise harms and maximise the benefits of e-cigarette use in prisons.

4. Suggest displaying table 3 after the “Misuse of e-cigarette among PiC to take illegal drugs”, since “NPS” was first appeared in this paragraph.

Our response: We have now defined NPS on page 11 given this is the first time the abbreviation is used, as per Reviewer 2’s suggestion below. On this basis we have retained the original positioning of Table 3.

Response to Reviewer #2 (comments received in PDF attachment):

1. Page 3, line 23: Suggests replacing ‘once this had been allowed for several months’ with ‘post implementation of the smoke-free policy’.

Our response: Thank you for your suggestion. We have now revised this sentence, including a more specific indicator of the timing of the study post-implementation as follows:

‘This study focuses on key challenges associated with e-cigarette use in prisons, using data collected from prison staff once e-cigarettes had been allowed in a smoke-free environment for 6-9 months.’

2. Page 4, line 47 – Is tobacco use totally prohibited by PiC?

Our response: We have now clarified in the third sentence of the introduction that tobacco stopped being sold in prisons ~two weeks before the smokefree policy was introduced. Yes, tobacco use is therefore totally prohibited by people in custody.

3. Page 4, line 59 – What is the referent?

Our response: We have now clarified this sentence, adding ‘*The introduction of e-cigarettes in prisons in Scotland*’ at the beginning.

4. Page 5, line 8 and 16 – suggests replacing e-cigarettes with ‘e-cigarette use’.

Our response: Many thanks for your suggestion, which we have incorporated in lines 8 and 16.

5. Page 5 line 39 – Is there some sort of screening process so that non-smoking PIC are not provided/cannot purchase e-cigarettes?

Our response: There is no screening process involved, and e-cigarettes are available to purchase by all people in custody in Scottish prisons aged 18+ via the prison canteen (shop).

6. Page 6, line 12 – Could you elaborate on some of these implications?

Our response: We have elaborated on the perceived implications of e-cigarette use for long-term tobacco and nicotine cessation, providing the example of uncertainties about whether e-cigarette use in prison increases or decreases the likelihood of return to smoking on release from prison.

7. Page 11, line 44 – Spell out NPS on first use

8. Page 13, line 29 – suggests replacing ‘on’ the halls with ‘in’ the halls

9. Page 17, line 9 – suggests adding ‘and short-term’ to the following sentence: ‘An area of concern to participants was the potential health impacts of (long-term) exposure to e-cigarette vapour at work.’

Our response: Many thanks for suggestions 7, 8 and 9, which are incorporated in the revised version of the manuscript.

10. Page 18, line 15 – Can any recommendations be given here?

Our response: We feel it is outwith our expertise to provide recommendations on the choice and design of rechargeable e-cigarette devices to reduce tampering, especially given rapid changes in the prison context, some of which may be confidential to those responsible for security within (particular) prisons. On this basis we have rephrased the sentence on line 15 to read: *‘These findings highlight the ongoing challenges for prison authorities to find appropriate e-cigarette devices for use in prisons, and other secure settings such as in-patient mental health services.’*

11. Page 19, line 17 – Before the smoke-free policy was implemented, where and when could tobacco smokers smoke? Were there restrictions on where and when? Could these be replicated for e-cigarette instead of being given unlimited access in their cells?

Our response: We have now clarified in the second sentence of the Introduction that the places where e-cigarette use is permitted mirrors the places where PiC were allowed to smoke tobacco prior to implementation of the smokefree policy.

Page 19, line 31 - What constitutes effective use?

Our response: We have revised this sentence and now refer to ‘appropriate’ rather than ‘effective’ use.

12. Page 19, line 38 – Did any of the participating prisons implement smoking/e-cigarette prevention/quit programs?

Our response: We have added a reference [2] to the Introduction signposting readers to detailed information about prison smoking cessations in Scotland.

Since these data were collected, NHS Health Scotland has developed guidance for practitioners supporting people who wish to cut down or quit vaping in prisons, which is mentioned in the Discussion on page 19. Plans to evaluate this guidance have been disrupted by the covid-19 pandemic.

VERSION 2 – REVIEW

REVIEWER	Lin Xiao Chinese Center for Disease Control and Prevention, Tobacco Control Office
REVIEW RETURNED	22-Oct-2021

GENERAL COMMENTS	On line 56-58 of page 15, you mentioned some solutions to the perceived challenges of vaping in prisons, such as "only selling single use e-cigarettes in prisons to limit product misuse, and supporting PiC to manage and, if desired, reduce use of rechargeable e-cigarettes". The data was showed in Table 4c-1. I am little bit confused how can you read these things from table 4c-1. Could you check it? Or could you find other data from your qualitative study records to support your results?
---

REVIEWER	Kahlia McCausland Curtin University - Perth City Campus, Collaboration for Evidence, Research and Impact in Public Health, School of Public Health
REVIEW RETURNED	25-Oct-2021

GENERAL COMMENTS	Thank you for addressing the first round of comments raised. I have added a couple more comments in the PDF I have attached, mainly in regards to some areas of your results that could be further fleshed
--

	out and areas that I believe are very important and need further attention in your discussion. The reviewer provided a marked copy with additional comments. Please contact the publisher for full details.
--	---

VERSION 2 – AUTHOR RESPONSE

Response to Reviewer #1:

1) On line 56-58 of page 15, you mentioned some solutions to the perceived challenges of vaping in prisons, such as "only selling single use e-cigarettes in prisons to limit product misuse, and supporting PiC to manage and, if desired, reduce use of rechargeable e-cigarettes". The data was showed in Table 4c-1. I am little bit confused how can you read these things from table 4c-1. Could you check it? Or could you find other data from your qualitative study records to support your results?

Our response:

Thank you for highlighting this point. We have amended Table 4c to include two new quotes from our data which better illustrate the possible solutions that staff raised here.

Response to Reviewer #2:

Thank you for addressing the first round of comments raised. I have added a couple more comments in the PDF I have attached, mainly in regards to some areas of your results that could be further fleshed out and areas that I believe are very important and need further attention in your discussion.

Page 5:

1) It is important to further discuss why rechargeable e-cigs are permitted versus other types? (Line 10)

Our response:

Thank you for your helpful comment. We have clarified the two types of e-cigarettes that are sold in prisons in Scotland - single-use and rechargeable – which reflects the types sold in wider society.

2) What is the reference here? (Line 12)

Our response:

We have amended the text in this sentence in response to comment 3 below, and on this basis no reference is required.

3) Which were? Or is that in cells and outdoor spaces as stated in the previous sentence? If so, perhaps consider putting that information here rather than beforehand (Line 15)

Our response:

We have reworded this sentence for clarity and to address point 3 above. It now reads: *'The places where e-cigarette use is permitted mirrors the places where PiC were allowed to smoke tobacco prior to implementation of the smoke-free policy (i.e. in designated rooms and in some outdoor spaces).'*

4) This whole paragraph is lacking citations (Line 43)

Our response:

Thank you for highlighting this. We have now added additional references after the sentence *'Current research evidence on e-cigarette use in prisons comes from two studies of tobacco and vaping in Scottish prisons (the Tobacco in Prisons study [TIPs] and the E-cigarettes in Prisons study.'*

5) tobacco and e-cigarette use? (Line 43)

Our response:

Thank you for your suggestion. In line 21, we note that from this point e-cigarette use will be referred to as vaping, so we have retained use of this term here on that basis.

Page 6:

6) I think this is a very important point that needs to be discussed further in your discussion and stems from my previous comments in the first round of review about limiting access to non-smokers/vapers (line 7).

Our response:

We have now added text to the discussion to clarify that prison staff did not raise concerns about use of e-cigarettes by the minority of never smokers in the prison population, but that ongoing monitoring would be helpful in tracking this issue in the future, so that swift corrective action could be taken if required.

7) This sentence needs to be revised and missing end bracket (line 32)

Our response:

We have deleted the bracketed text and clarified in the remaining sentence that both prison staff and some people in custody (PIC) voiced important issues and challenges in relation to permitting PIC to vape.

8) prison staff? (Line 55 and again at Line 60)

Our response:

We have amended text in both lines as per your helpful suggestion.

Page 7

9) challenges of permitting the use of e-cigarettes in prisons? Or something to that effect (line 4)

Page 8

10) at the prison? (line 57)

Page 9

11) the prison the participant worked at, their interview ID number and smoking and vaping status (line 9)

Page 10

12) challenges of permitting the use of e-cigarettes in prisons? Or something to that effect (line 9)

13) e-cigarette availability in prison (line 14)

Our response:

We have amended the text on pages 7, 8, 9 and 10 as per your helpful suggestions.

14) supported how? (line 21)

Our response:

We have clarified that e-cigarettes were generally believed to have supported smoke-free prison policy implementation by helping PIC to manage without tobacco.

15) This last sentence could be unpacked further - don't leave it up to the reader to make interpretations of the data (line 28)

Our response:

Thank you for your helpful suggestion. We have added an additional sentence to expand on this point, which reads: *'For example, some staff suggested in retrospect that the service might have been able to manage any temporary disruption that could have arisen from the removal of tobacco from prisons.'*

Page 13

16) What are residential staff? Perhaps a footnote explanation would be helpful? (line 23)

Our response:

We have clarified this by amending the start of this sentence, which now reads: *'Staff working in residential areas of prisons...'*

17) I think the issues raised in this para are very important points to address in your discussion, if not already.(Line 32)

Our response:

We agree that these are important issues, and they are addressed in our discussion from p18 (line 23) to p19 (line 6).

18) How? (Line 31)

Our response:

We have added the following text to illustrate an example of the ways in which staff perceived NPS use in prisons had changed their work environment, '*for example by undermining safety and security for staff and PiC.*'

Page 14

19) Can you provide the cost for e-cig products vs tobacco? (line 19)

Our response:

Pricing structures for products for sale on the prison canteen list change over time. There was only a very short period when both tobacco and rechargeable e-cigarette products were available for purchase to people in custody prior to the introduction of smoke-free policy. We do not have data on current pricing and do not think that historical data would be a helpful addition to the paper.

Page 15

20) COVID (line 55)

Our response:

We have amended this text in line with your suggestion.

Page 16

21) which are? (line 10)

Our response:

We have clarified the meaning of 'prohibited areas' by adding the following text: '*(e.g. vaping outside designated rooms and some outdoor spaces)*'

22) I think this is another very important point to address in your discussion, if not already. (line 39)

Our response:

We agree that this is another important issue, which has been addressed in our discussion on p20, from line 15. We have also added text to update this section of text.

23) Is it the use of e-cigarettes or nicotine addiction? (line 59)

Our response:

Staff perceived both to be important. On this basis we have added text to convey that staff perceived the sole availability of higher strength e-cigarettes in prisons '*as a potential significant barrier to enabling PiC to reduce use of e-cigarettes/nicotine dependence.*'

Page 17

24) 'at a symbolic and/or practical level' - can this be further explained? (line 60)

Our response:

We have removed this text as we feel it is clear in the rest of the sentence why staff might find it hard that policies on e-cigarette use are more lax in prisons than elsewhere.

Page 18

25) I don't think this acronym [SHS] has been used before (line 4)

Our response:

Thank you for highlighting this. We have now written 'second-hand smoke' in full as this is the first and only time it is used in the main text of the paper.